# Biologically Plausible Neural Networks via Evolutionary Dynamics and Dopaminergic Plasticity

**Sruthi Gorantla**
National University of Singapore
sruthi@comp.nus.edu.sg

**Anand Louis**
Indian Institute of Science
anandl@iisc.ac.in

**Christos H. Papadimitriou**
Columbia University
christos@columbia.edu

**Santosh S. Vempala**
Georgia Tech.
vempala@gatech.edu

**Naganand Yadati**
Indian Institute of Science
y.naganand@gmail.com

## Abstract

Artificial neural networks (ANNs) lack in biological plausibility, chiefly because backpropagation requires a variant of plasticity (precise changes of the synaptic weights informed by neural events that occur downstream in the neural circuit) that is profoundly incompatible with the current understanding of the animal brain. Here we propose that backpropagation can happen in evolutionary time, instead of lifetime, in what we call neural net evolution (NNE). In NNE the weights of the links of the neural net are sparse linear functions of the animal's genes, where each gene has two alleles, 0 and 1. In each generation, a population is generated at random based on current allele frequencies, and it is tested in the learning task through minibatches. The relative performance of the two alleles of each gene is determined, and the allele frequencies are updated via the standard population genetics equations for the weak selection regime. We prove that, under assumptions, NNE succeeds in learning simple labeling functions with high probability, and with polynomially many generations and individuals per generation. NNE is also tested on MNIST with encouraging results. Finally, we explore a further version of biologically plausible ANNs (replacing backprop) inspired by the recent discovery of dopaminergic plasticity.

## 1 Introduction

In his Turing award lecture [6], neural networks pioneer Geoff Hinton opined that *"evolution can't get gradients because a lot of what determines the relationship between the genotype and the phenotype is outside your control"*. We beg to differ. The well-established equations of population genetics governing evolution under recombination and weak selection [3, 4] do bear a similarity to gradient descent – and, even more, to no-regret learning [4]. In this paper, we show that, in fact, quite effective training of neural nets can be carried out without backpropagation, and in evolutionary time, through a variant of gradient descent. In other words, we argue both theoretically and through experiments that brain circuits specializing in classification tasks could have evolved.

The towering empirical success of ANNs has brought into focus their profound incongruity with what we know about the brain: backpropagation requires that plasticity be informed by downstream events. Clever versions of ANNs have been proposed recently that avoid this criticism: ANNs whose backward weights are random and fixed [9], a backpropagation interpretation of STDP (a widely accepted theory of plasticity) [1], or ANNs driven by neural competition [8].

Here we take a very different approach. We believe that, while forward neural computation is coterminous with life, backpropagation (i.e., local feedback on the performance of the circuit) can

be effectively carried out over evolutionary time. Suppose that the brain circuitry for a particular classification task, such as "food/not food", is encoded in the animal's genes, assuming each gene to have two alleles 0 and 1. A (haploid) genotype is a bit string, and the weight of each link of the neural network (assumed to be fixed for simplicity) is a sparse linear function of the genes. Evolution proceeds in generations. At each generation, a gene is an independent binary variable with fixed probability of 1. A population is sampled from this distribution of genotypes, and it experiences a sequence of inputs to the brain circuit. Fitness of each genotype depends, to some small degree, on the animal's success over its lifetime in the specific classification task. In the next generation, the allele frequencies will change slightly, depending on how each allele of each gene fared cumulatively (over both all inputs and all genotypes containing it) in the classification task. These changes follow the standard population genetics equations for the weak selection regime, see [3, 4]; weak selection means that the classification task is only one of the many biological functions (digestion, locomotion, etc.) that affect the animal's fitness.

The question is, can competent brain circuits evolve this way? We offer both theoretical and experimental evidence that this is indeed the case.[1] Our experiments are on the MNIST data set, where we apply Neural Network Evolution (NNE) to learn to categorize digits. In fact, we use the simplest possible version corresponding to a single hidden layer of activations. We find that this already gives surprisingly good accuracy rates. We then use a simple model of arbitrary linear *target* functions and show that NNE will converge to the target. We show in Section 2 that NNE shadows the gradient of the squared error function over a mini-batch, and converges to a $0/1$ allele distribution.

We also propose a totally different alternative to backprop — biologically plausible ANNs based on dopaminergic plasticity. It was recently established experimentally [14] that weights in certain synapses (from the cortex to the striatum but not only) are increased if dopamine was released within 0.5-2 seconds after the synapse's firing. Inspired by this experiment, we define dopaminergic neural nets (DNN), in which the weight of a link that fired (that is, both nodes fired during the current minibatch) is modified by a multiple of $(\frac{1}{4} - err^2)$, where $err$ is the error of the current minibatch. That is, links that fired are rewarded if the result was good, and punished if it was not. We show experimentally that such ANNs can also learn to classify quite well.

**Our Contributions.** In Section 2, we study a 1-layer NNE with squared loss as loss function. In Section 3, we discuss preliminary experiments showing that NNE performs reasonably well on MNIST.

## 2 A rigorous analysis for the case of learning linear functions

A genotype can be viewed as a vector $x \in \{0, 1\}^n$. A probability distribution over the genotypes is given by a vector $p \in [0, 1]^n$; a genotype $x$ is sampled by setting $x(i) = 1$ with probability $p(i)$, independently for each $i$. The neural network corresponding to a genotype $x$ is a *feed-forward neural network* (FFNN) whose weights are computed as follows. For a prediction network having $m$ links, the weights of the links are given by $Wx$, where $W$ is an $m \times n$ sparse *weight generation matrix*. We choose the entries of $W$ to be random and i.i.d.: with probability $\beta$, $W(i, j)$ is chosen uniformly at random from $[-1, 1]$, and is 0 with probability $1 - \beta$.

The input to the network is a vector $y$ drawn from a distribution $\mathcal{D}$ and has a label (possibly real-valued) $\ell(y)$. The output of the network on an input $y$ is $\mathsf{NNE}_x(y)$. In the simplest linear case, $y \in \mathcal{R}^m$ and $\mathsf{NNE}_x(y) = x^T W^T y$. In our experiments (Section 3), we study the case when $\mathcal{D}$ is the uniform distribution over MNIST, and $\mathsf{NNE}_x(\cdot)$ is a 1-layer neural network with a ReLU output gate (see Section 3 for formal definition).

For each genotype $x$, we measure its performance by computing the loss $L(\mathsf{NNE}_x(y), l(y))$ (this could be squared loss, cross-entropy loss, etc.). For a probability distribution over genotypes $p$, we define the loss as

$$\mathcal{L}(p) := \mathsf{E}_{x \sim p} \mathsf{E}_{y \sim D} L(\mathsf{NNE}_x(y), \ell(y)).$$

---

[1]We note incidentally that NNE is very distinct from neuroevolution (see the recent survey [12]), which optimizes ANN architecture and hyperparameters through genetic algorithms

We calculate the rewards $f^t(i)$ and $\bar{f}^t(i)$ as the expected negative loss whenever the allele is present and absent respectively.

$$f^t(i) = \mathsf{E}_{x \sim p^t} \left[ \mathsf{E}_{y \sim D} \left[ -L(\mathsf{NNE}_x(y), \ell(y)) \right] \mid x(i) = 1 \right]. \tag{1}$$

and

$$\bar{f}^t(i) = \mathsf{E}_{x \sim p^t} \left[ \mathsf{E}_{y \sim D} \left[ -L(\mathsf{NNE}_x(y), \ell(y)) \right] \mid x(i) = 0 \right]. \tag{2}$$

For the next generation we calculate,

$$p = p^t(i)(1 + \epsilon f^t(i)) \qquad \text{and} \qquad q = (1 - p^t(i))(1 + \epsilon \bar{f}^t(i))$$

We normalize $p$ and $q$ to make it a probability distribution. Thus the allele probabilities for the next generation will be,

$$p^{t+1}(i) = \frac{p}{p+q} = \frac{p^t(i)(1 + \epsilon f^t(i))}{1 + \epsilon \bar{f}^t(i) + \epsilon p^t(i)(f^t(i) - \bar{f}^t(i))}. \tag{3}$$

This is the standard update rule in population genetics under the weak selection assumption. The multiplier $\epsilon$ captures the small degree to which the performance of this task by the animal confers an evolutionary advantage leading to larger progeny.

Our first observation is that perfomance per allele is in fact a function of the gradient of the loss function.

**Lemma 1**

$$\mathcal{L}(p^t) = -\bar{f}^t(i) - p^t(i)(f^t(i) - \bar{f}^t(i)) \quad and \quad \frac{\partial}{\partial p^t(i)} \left( \mathcal{L}(p^t) \right) = -(f^t(i) - \bar{f}^t(i)).$$

We use this to prove the following theorem.

**Theorem 1** *Fix $\delta > 0$. Suppose $\nabla^2 \mathcal{L}(z) \preceq H \cdot I \; \forall z \in [0,1]^n$. Let $U := \sup_{p \in [0,1]^n} \mathcal{L}(p)$ and $S_t := \{i \in [n] | \delta \le p^t(i) \le 1 - \delta\}$. For $\epsilon \le \min\{1/(\max\{2U, 1\}), 2/H, 1\}$, there is an $\eta > 0$ s.t.*

$$\mathsf{E}(\mathcal{L}(p^{t+1})) \le \mathcal{L}(p^t) - \eta \sum_{i \in S_t} \left( \nabla_i \mathcal{L}(p^t) \right)^2.$$

## 2.1 Learning linear functions

In this section, we show that in the case of a linear target functions, with high probability, NNE converges to an allele distribution $p$ which is arbitrarily close to the correct linear labeling. Our NNE has $m$ input gates connected to one output gate (i.e., no hidden layers). For a genotype $x$, the weights of the connections are given by $Wx$. On input $y$, the NNE outputs $x^T W^T y$.

**Theorem 2** *Let $D$ be the uniform distribution over vectors in an $n$-dimensional unit ball. Let $a$ be a fixed vector with $\|a\| \le 1$, such that the label of $y$ is $\ell(y) := a^T y$. Let $W$ have i.i.d. entries with $W_{ij} = \pm\sqrt{m/d}$ with probability $d/m$ and $0$ with probability $1 - (d/m)$. Then, for any $\delta \in (0,1]$, with $n = O(m + \log(1/\delta))$, with probability at least $1 - \delta$, there exists an allele distribution $p$ s.t. $Wp = a$. Moreover, with probability at least $3/4$, for any $\epsilon \in (0,1]$, with $n = \Omega(m(\log(1/\epsilon)/\epsilon^2)$, there is an $x \in \{0,1\}^n$ s.t. $\frac{(Wx) \cdot a}{\|Wx\|\|a\|} \ge 1 - \epsilon$.*

We remark that the above guarantee works for *every* linear target function in $\mathbb{R}^m$. To learn, with high confidence, the target function from among $d$ unknown (arbitrary) linear functions, $m$ above can be replaced by $\log d$.

# 3 Experiments

## 3.1 NNE on MNIST

We study the effectiveness of NNE by evaluating its classification performance on the MNIST dataset.

To train an NNE via evolution of $T$ generations of genotypes, we fix a sufficiently large population size $\mathcal{N}$. Each generation $t \in [T]$ consists of a sample of $\mathcal{N}$ independently sampled genotypes from the allele distribution $p^t$, we denote this sample by $\mathcal{P}^t$. This distribution is updated based on the average performance $f^t(i)$ and $\bar{f}^t(i)$ of all the genotypes on a task, in our case, MNIST handwirtten digit recognition task. We let the allele distribution $p^t$ evolve over $T$ generations in this manner.

| model | MNIST 0 to 4 | MNIST 0 to 9 |
|---|---|---|
| NNE | 92.1 | 78.8 |
| NNE + SignSGD | 91.6 | 85.6 |
| SGD | $96 \pm 0.4$ | $88.2 \pm 0.75$ |

Table 1: Accuracy rates of NNE on MNIST test digits.

**Experimental setup.** We use 200 training samples for each of the digits, drawn uniformly at random from MNIST; we denote this set of training examples by $S$. $p^1$, the allele distribution for the first generation, is sampled uniformly at random from $[0,1]^n$. We evaluate the performance of the alleles over $\mathcal{N} = 1000$ genotypes.

Our network has 784 input units, one hidden layer of $|h^1| = 1000$ units with ReLU activation and an output layer of 10 units with softmax activation. We add a sparse random graph between the input layer and the hidden layer: between a neuron in the input layer and a neuron in the hidden layer, we independently add an edge with probability 0.1. The hidden layer is fully connected to the output layer. We choose $\beta = 0.0025$ for our experiments, i.e., each edge weight is a sparse random function of only $\beta$ fraction of the alleles. For the input sample $y$, $\ell(y)$ is now a one-hot encoding of the label, and $\mathsf{NNE}_x(y)$ is the soft-max output of the network. We use the cross-entropy loss function, $L(\mathsf{NNE}_x(y), \ell(y)) = -\sum_{c \in [C]} \ell(y)_c \log(\mathsf{NNE}_x(y)_c)$.

If a classifier were to randomly guess the label of an input intance, its loss function value would be $\alpha := -\log(1/10)$. We use the relative performance of the genotype w.r.t. to a random guess for our updates. To this end, we define for a genotype $x$, $\delta_x := \frac{1}{|S|} \sum_{s \in [S]} \max\{0, \alpha^2 - L(\mathsf{NNE}_x(y), \ell(y))^2\}$. For each allele, we calculate the rewards $f^t(i)$ and $\bar{f}^t(i)$ whenever the allele is present and absent respectively.

$$f^t(i) = \frac{\sum_{x \in \mathcal{P}^t} \delta_x x(i)}{\sum_{x \in \mathcal{P}^t} x(i)} \qquad \text{and} \qquad \bar{f}^t(i) = \frac{\sum_{x \in \mathcal{P}^t} \delta_x (1 - x(i))}{\sum_{x \in \mathcal{P}^t} (1 - x(i))}.$$

The allele distribution for the next generation is updated using equation 3.

NNE as described above achieves **78.8%** test accuracy on the full MNIST test set. While this is somewhat far from the state of art in classification of MNIST images, our results demonstrate that very basic NNEs can perform reasonably well in this task. See experimental results in Appendix showing the effect of number of genes on performance.

**NNE with output layer training.** The biological implausibility objection of using stochastic gradient based updates is less acute for the output layer, since in animal brains synaptic changes due to plasticity happen at the post-synaptic neuron, and for the output layer this is the output neuron. Even then, computing exact (or approximate) gradients is a nontrivial computational task; instead we consider using just the *sign* of the gradient for only the output layer as a lifetime training mechanism.

For the same network described as above, we randomly initialize the network weights using allele distribution learned using the NNE. We then calculate the sign of the gradient of the output layer weights and update the weights in the opposite direction (SignSGD), using a sufficiently small learning rate $\epsilon'$, similar to stochastic gradient descent. For $i$ in the hidden layer and $j$ in the output layer, the update is

$$w_{ij} := w_{ij} - \epsilon' \cdot sign\left((z_j - \ell(y)_j)h_i\right) \tag{4}$$

where $h_i$ is output of the neuron $i$, and $z_j$ softmax output of neuron $j$. SignSGD has been shown to be effective for traning large deep neural networks (for e.g., see [2]).

We perform a few hundred iterations of this training using batch size 50. In this experiment (NNE + SignSGD), we obtain **86.3%** accuracy on full MNIST test set. This further demonstrates that biologically plausible neural networks can perform reasonably well in this task.

Table 1 compares the results of all the models along with the baseline, stochastic gradient descent trained on the same subset of MNIST.

| $h$ | SGD | SignSGD | DNN |
|---|---|---|---|
| 1000 | 90.34 | 86.84 | 84.84 |
| 100,000 | 92.56 | 90.21 | 90.76 |

Table 2: Test accuracies of different models for different $h$.

## 3.2 Dopaminergic Neural Nets (DNNs)

DNNs are biologically plausible ANNs based on dopaminergic plasticity. They learn by a weak form of immediate reinforcement - "rewarding" synapses whose firing led to a favourable outcome. If a connection between two neurons has fired during a training step, then its weight is increased if the square error was low (less than $\frac{1}{4}$). In this section, we demonstrate that simple DNNs can perform reasonably well for tasks like classifying the images in the MNIST dataset.

**Experimental setup.** For our experiments we use a network consisting of an input layer, a single hidden layer, and an output layer consisting of 784, $h$, and 10 neurons respectively. Each neuron in the input layer has a link to each neuron in the hidden layer, and its weight is initialised by the popularly used Kaiming Uniform (more commonly called He initialisation [5]). These weights are unchanged through out the learning process. Recent theoretical results suggest that a large enough random layer is sufficiently rich and efficiently trainable [13] (see also [11]).

Each neuron in the hidden layer has a link to each neuron in the output layer. The output layer outputs the softmax score. The weights of this layer are learned using plasticity based updates. On seeing an input $y$, the DNN tries to predict the label of $y$; let us denote this by $\mathsf{DNN}_W(y)$. If the DNN got the prediction correct, i.e. the loss $L(\mathsf{DNN}_W(y), \ell(y))$ is at most $\epsilon_0$, then weight $w_{ij}$ get increased by a small amount, provided the output neuron $j$ has low error (i.e. $|z_j - l(y)_j|^2 \le 1/4$) where $z_j$ is the $j$th coordinate of $\mathsf{DNN}_W(s)$.

Formally, the update rule is as follows for $i$ in the hidden layer and $j$ in the output layer.

$$ w_{ij} = w_{ij} + \epsilon_1 \frac{\max\left\{0, \frac{1}{4} - |z_j - \ell(y)_j|^2\right\} \cdot \max\left\{0, L(\mathsf{DNN}_W(y), \ell(y)) - \epsilon_0\right\}}{\left(\frac{1}{4} - |z_j - \ell(y)_j|^2\right) \cdot (L(\mathsf{DNN}_W(y), \ell(y)) - \epsilon_0)}. $$

**Experimental results.** To study the effectiveness of our DNN in the 10-class MNIST digit classification, we compare its peformance with some other standard baselines.

1. SGD: In this we use the standard stochastic gradient descent (with the Adam optimiser [7]) based updates to train our network.
2. SignSGD: As before, we use the sign of the gradient for updates (equation 4).

Table 2 shows the results for different $h$ values. All results are after 500 epochs of training. As with NNE we use the cross-entropy loss for all the models. We found that $\epsilon_0 = 0.75$ and $\epsilon_1 = 1$ for the DNN gives reasonable performance. Our DNN gives encouraging results and is comparable to SignSGD in performance.

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

## Acknowledgements

We would like to acknowledge support for this project. CP was supported by NSF grant CCF-1763970. SV was supported in part by NSF awards CCF-1563838, CCF-1717349 and CCF-1909756. AL was supported in part by SERB Award ECR/2017/003296 and a Pratiksha Trust Young Investigator Award.

## Appendix

**Number of genes.** A crucial choice for an NNE is the number of genes. In our experiments, we use a few thousand genes; this is not unreasonable as it is estimated that about $5,000$ genes are expressed in the cells of the mammalian brain. To investigate further, we compare the performance of our algorithm with increasing values of $n$ (the number of genes). Figure 1 presents the validation accuracy trends on the same network described above for five class $[0-4]$ classification and for full MNIST dataset. We observe that the accuracy rate of the network improves significantly with increase in the number of genes. However, it requires much longer training time to achieve a desired accuracy rate.

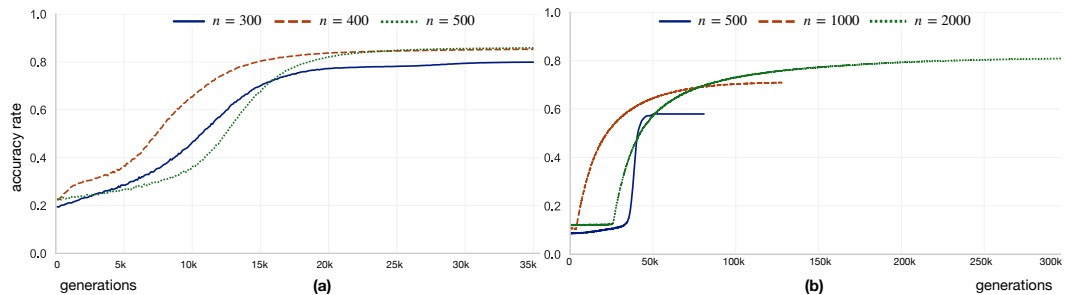

Figure 1: Number of genes ($n$) vs performance of NNE: **(a)** Accuracy rates of NNE on MNIST $0-4$ showing the effect of number of genes on performance. **(b)** Similarly, we also plot the accuracy rates of NNE on MNIST $0-9$ dataset while varying the number of genes. The accuracy trends show that more the number of genes, better the performance of NNE, but at the cost of more training time.

**Convergence of allele distributions.** We repeat training NNE for many (hundreds of thousands) generations. As our theoretical results predict (see also [10]), the vast majority of genes have allele probabilities that are very close to 0 and 1. Figure 2 shows the fraction of allele probabilities that are at a distance $[x, 1-x]$ from 0 or 1, i.e., $y$ is calculated as $y = 1 - \frac{|\{i:\min\{p^t(i),1-p^t(i)\}\leq x\}|}{n}$.

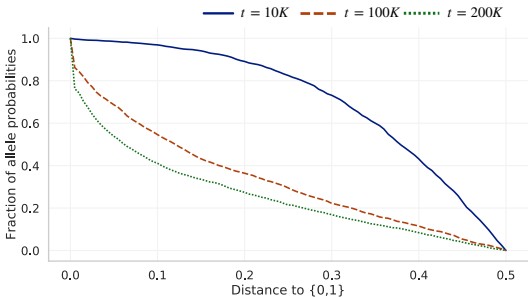

Figure 2: Convergence of allele distribution after $t$ generations: x-axis shows the distance of the allele probabilities from 0 or 1 and the y-axis shows the fraction of $n$ allele probabilities that are at a distance $[x, 1-x]$ from 0 or 1.

