# OpenReview forum: "Biologically Plausible Neural Networks via Evolutionary Dynamics and Dopaminergic Plasticity"
_NeurIPS.cc/2019/Workshop/Neuro_AI — Real Neurons & Hidden Units @ NeurIPS 2019 Poster_

### Official Review · AnonReviewer1 · 2019-09-20
**Interesting line of work, but not convincing enough results and not clear enough**

**Clarity:** 2

**Comment:**

Definitely an interesting line of work. But the results need to be presented more clearly to really evaluate the worth of this particular model.

Some demonstration that it does indeed work according to the theory provided (e.g. on simpler regression problems) would be useful to get the idea off the ground.

The title and results are also a combination of two ideas (the evolutionary algorithm, and the sign matched 'dopamine' inspired updates). I would focus on one idea at a time or explain how they do go together

**Category:**

Common question to both AI & Neuro

**Clarity Comment:**

There are many small points and omitted details that make the presented work hard to evaluate. For instance:

- Is the beta at line 67 the same as 1-gamma at line 119?
- Line 53: by ‘0/1 allele distribution’ you mean a deterministic distribution?
- I’m confused about the set S. Is it a small set of 2000 elements of MNIST, sampled uniformly from all of MNIST, or only from digits 0-4? The text seems to suggest S is just 2000 elements from MNIST, while figure 1 presents results from digits 0-4 or the full set. Citing training results of 83% on S and 79% test on full MNIST, in the text, are both referring to Fig 1b? It took a number of passes through the text to figure out what was being plotted in relation to the analysis that was done.
- The model could be more explicitly defined, though I know space is limited in this submission


**Evaluation:**

2: Poor

**Importance:**

3: Important

**Importance Comment:**

It is interesting to think about how evolution interacts with learning that takes place during an organism’s lifetime. While there is fruitful work to be done here, the motivation made by the authors needs a little more work. For instance, what types of learning do we expect to be encoded in an animal’s genes as opposed to its acquired synapses? Claiming that it’s more ‘biologically plausible’ is not good enough: there are many plausible models that do not resort to evolution over generations.

**Intersection:**

4: High

**Intersection Comment:**

There is room here for these type of models to benefit both theoretical neuroscience, and AI. If something shows more promise on something like MNIST it may be of benefit to AI. If some of the details about how an evolutionary algorithm interacts with ‘within-lifetime’ learning then it could benefit neuro.

**Rigor Comment:**

The fact that in Fig 1b it takes a while for the n=2000 model to learn anything suggests there may be significant variability in the results when repeated many times. Some repeated runs of the algorithm for a given number of genes would be helpful.

The addition of the dopaminergic neural nets is not well enough explained and introduced to warrant inclusion. It appears just as a random addition to the model. It’s not clear how the specific dopamine-related timing result they mention is incorporated in their model. More generally, reward-modulated plasticity is very well explored, why not just use these results?

A result of 83% test accuracy with a non-linear network on MNIST is not so encouraging… given that linear networks can perform better. Some simpler task might be worth investigating to get a better intuition for what in this model works and what doesn’t, before trying MNIST.


**Technical Rigor:**

2: Marginally convincing

---

### Official Review · AnonReviewer2 · 2019-09-24
**Neural networks can be instantiated and trained in evolutionary-time as sparse linear functions of genes**

**Clarity:** 3

**Category:**

Common question to both AI & Neuro

**Clarity Comment:**

The approach is certainly introduced in interesting way and the methods are reasonably easy to follow.

**Evaluation:**

2: Poor

**Importance:**

3: Important

**Importance Comment:**

The largest motivation seems to be the biological implausibility of backpropagation. However, many studies have shown that all aspects of backprop can be, and most likely are, realized in biological networks (error calculation, weight transport, etc. - e.g., Lillicrap & Santoro, 2019; Akrout et al. 2019). Therefore, this motivation is not enough alone.

**Intersection:**

4: High

**Intersection Comment:**

To what extent human abilities are represented at the genomic level vs. learned within a lifetime is certainly an interesting biological question, but it’s applicability to machine learning has yet to be shown convincingly.

**Rigor Comment:**

While the technical proofs are useful, the performance on MNIST is not particularly convincing. Additionally, the results seem quite noisy and would benefit from many repeated runs.

What would be more interesting is to address how learning could take place through a combination of evolution- and life-time mechanisms, as opposed to a purely evolutionary-time mechanism.

The inclusion of learning based on dopaminergic plasticity seems quite arbitrary. Additionally, the authors cite one biological paper, but methods like this have been used for decades in one way or another (e.g. https://openreview.net/forum?id=r1lrAiA5Ym)

**Technical Rigor:**

2: Marginally convincing

---

### Official Review · AnonReviewer3 · 2019-09-27

**Clarity:** 3

**Comment:**

I did not find the proposal that individual synapses are learned via evolution to be compelling. This claim is seemingly contradicted by strong experimental evidence of learning at all scales during animals lifespans, and by an observation of the number of bits in the genome vs. the number of synapses in the brain. It would require stronger evidence, and discussion of the potential barriers, for me to take this proposal more seriously.

**Category:**

Common question to both AI & Neuro

**Clarity Comment:**

I understood the core idea, but felt that the idea itself had not been carefully thought through.


**Evaluation:**

2: Poor

**Importance:**

2: Marginally important

**Importance Comment:**

I don't believe this paper makes a compelling enough argument to cause readers to rethink what is learned in evolutionary rather than developmental time.


**Intersection:**

3: Medium

**Intersection Comment:**

This was mainly a proposal for evolutionary learning of biological network weights. The idea was tested using an artificial neural network, but was not otherwise strongly connected to machine learning.

**Rigor Comment:**

I believe the algorithm functions as proposed. I suspect it is not a reasonable model for learning genetic influence on synapses.

**Technical Rigor:**

3: Convincing

---

### Decision · Program_Chairs · 2019-10-02

Accept (Poster)